# Giant extinct caiman breaks constraint on the axial skeleton of extant crocodylians

**Torsten M Scheyer[1]\*, John R Hutchinson[2], Olivier Strauss[1], Massimo Delfino[3,4], Jorge D Carrillo-Briceño[1], Rodolfo Sánchez[5], Marcelo R Sánchez-Villagra[1]**

[1]University of Zurich, Palaeontological Institute and Museum, Zurich, Switzerland; [2]Structure & Motion Laboratory, Department of Comparative Biomedical Sciences, The Royal Veterinary College, Hatfield, United Kingdom; [3]Dipartimento di Scienze della Terra, Università di Torino, Torino, Italy; [4]Institut Català de Paleontologia Miquel Crusafont, Universitat Autònoma de Barcelona, Barcelona, Spain; [5]Museo Paleontológico de Urumaco, Urumaco, Venezuela

**Abstract** The number of precaudal vertebrae in all extant crocodylians is remarkably conservative, with nine cervicals, 15 dorsals and two sacrals, a pattern present also in their closest extinct relatives. The consistent vertebral count indicates a tight control of axial patterning by *Hox* genes during development. Here we report on a deviation from this pattern based on an associated skeleton of the giant caimanine *Purussaurus*, a member of crown Crocodylia, and several other specimens from the Neogene of the northern neotropics. *P. mirandai* is the first crown-crocodylian to have three sacrals, two true sacral vertebrae and one non-pathological and functional dorsosacral, to articulate with the ilium (pelvis). The giant body size of this caiman relates to locomotory and postural changes. The iliosacral configuration, a more vertically oriented pectoral girdle, and low torsion of the femoral head relative to the condyles are hypothesized specializations for more upright limb orientation or weight support.

## Introduction

The Neogene crocodylian fauna of northern South America is remarkable in terms of species richness, levels of species sympatry, and ecomorphological specialization (e.g., *Riff et al., 2010*; *Sánchez-Villagra and Aguilera, 2006*; *Scheyer et al., 2013*; *Scheyer and Delfino, 2016*). Among a plethora of taxa, *Purussaurus* was an exceptionally large caiman (Alligatoroidea) that lived in the northern neotropics of South America in the middle and late Miocene (ca. 13–5 Ma). Its impressively large skull is the basis of its taxonomy, which encompasses three species distributed in localities in Brazil, Peru, Colombia and Venezuela (*Aguilera et al., 2006*; *Langston, 1965*; *Mook, 1921a*; *Mook, 1942*; *Riff et al., 2010*; *Salas-Gismondi et al., 2015*). The postcranial anatomy of *Purussaurus*, as that of most extinct crocodylians, is still poorly known. We report here on the discovery of an exceptional skeleton of *Purussaurus mirandai* and several other remains from the late Miocene in Venezuela. Archosauria are represented today by 10,000 + species of birds (crown Aves), but less than 30 species of crocodylians (crown Crocodylia). As sister taxa, both groups have undergone profound changes in their body plans throughout ca. 250 million years of evolutionary history. In comparing the morphology of extant Archosauria, birds reveal much larger variation in body shape, musculoskeletal form and function, ecology, and lifestyle; whereas crocodylian species resemble each other much more closely due to their shared amphibious and overall carnivorous lifestyle. When examining the underlying developmental patterns in archosaurs, the difference in variation is also reflected in the axial patterning of the vertebral column in Aves and Crocodylia, with the former having variable precaudal vertebral numbers and the latter showing a conserved pattern (*Mansfield and Abzhanov, 2010*;

**\*For correspondence:**
tscheyer@pim.uzh.ch

**Competing interests:** The authors declare that no competing interests exist.

**eLife digest** All living crocodiles, alligators, caimans and gharials – collectively called crocodylians – have a similar body plan that suits their lifestyles as aquatic ambush predators. This similarity extends throughout their bodies, including the skeleton. Their backbones, for example, always have nine vertebrae in the neck, 15 in the trunk, and two in the pelvis. Closely related extinct crocodylians also organize their spines in the same way.

Scheyer et al., however, now report that one extinct caiman called *Purussaurus mirandai* had a spine that was structured unlike any other known crocodylian. Weighing almost three tons (~2,600 kg), the 10-meter-long *Purussaurus* was more than twice as heavy as the largest living crocodylian, the saltwater crocodile. When Scheyer et al. examined fossilized remains from Venezuela that are estimated to be between 7–5 million years old, they found an extra vertebra in the creature's pelvic area and one less in its trunk. Scheyer et al. speculate that this unusual arrangement may have helped the extinct creature to support its massive weight and compensate for the strain imposed on its skeleton.

Within the animal kingdom, so-called homeobox genes dictate how different body structures, including the spine, develop in embryos. Shifts in where these genes are active in the embryo can result in an extra pelvic vertebra in humans and other animals. Scheyer et al. conclude that changes in the boundaries of the activity of homeobox genes may also explain the extra pelvic vertebra in this ancient caiman.

It is not yet clear if other extinct crocodylians had extra pelvic vertebrae as well. But these new findings are likely to lead to more research on related giant crocodylian fossils to find out. Such research could help scientists to better understand the biomechanics of crocodylians and may lead to new insights on caimans, which have thrived in the tropics of northern South America for the past seven million years. Further research in this area may also help explain how these reptiles have adapted to their environment and the role they play in their ecosystems, which is currently threatened by human activity.

*Müller et al., 2010*). Finally, extant crocodylians show distinctly low genome-wide evolutionary rates compared to those of birds, which could be linked to prolonged generation times in the former clade (*Green et al., 2014*). These low evolutionary rates could potentially underlie the generally lower morphofunctional disparity seen in the post-Cretaceous crocodylian body plans (*Brusatte et al., 2010*; *Stubbs et al., 2013*).

Vertebrate axial patterning by means of *Homeobox* (*Hox*) gene expression has been extensively studied in model organisms, including the chick, since the 1980 s (see *Favier and Dollé, 1997*; *McGinnis and Krumlauf, 1992*; *Wellik, 2007* for overviews). The study of the developmental patterning and associated gene expressions in crocodylians (among other extant reptiles), on the other hand, has only recently received attention (*Böhmer, 2013*; *Böhmer et al., 2015*; *Mansfield and Abzhanov, 2010*), with a focus on the presacral patterning of the body. These developmental studies as well as comparative anatomy (*Hoffstetter and Gasc, 1969*; *Mook, 1921b*) corroborate the general precaudal count of all crown Crocodylia to consist of nine cervicals, 15 dorsals (thoracic and lumbar) and two sacrals. Various pathological conditions have been reported using classical dissection (e.g., *Baur, 1886*; *Baur, 1889*; *Reinhardt, 1873*; *Reinhardt, 1874*).

Here, we present all relevant axial and appendicular material of the extinct giant caimanine *Purussaurus mirandai*, including a revised character scoring and phylogenetic analysis for the species, as the first non-pathological case within crown-Crocodylia that deviates from the highly conserved precaudal count of the group. Comparison with pathological (e.g., the presence of congenital malformation lumbosacral transitional vertebrae) and non-pathological extant crocodylians served as the basis for elucidating developmental patterns for the sacralisation of the last dorsal (i.e., lumbar) vertebra in the extinct species.

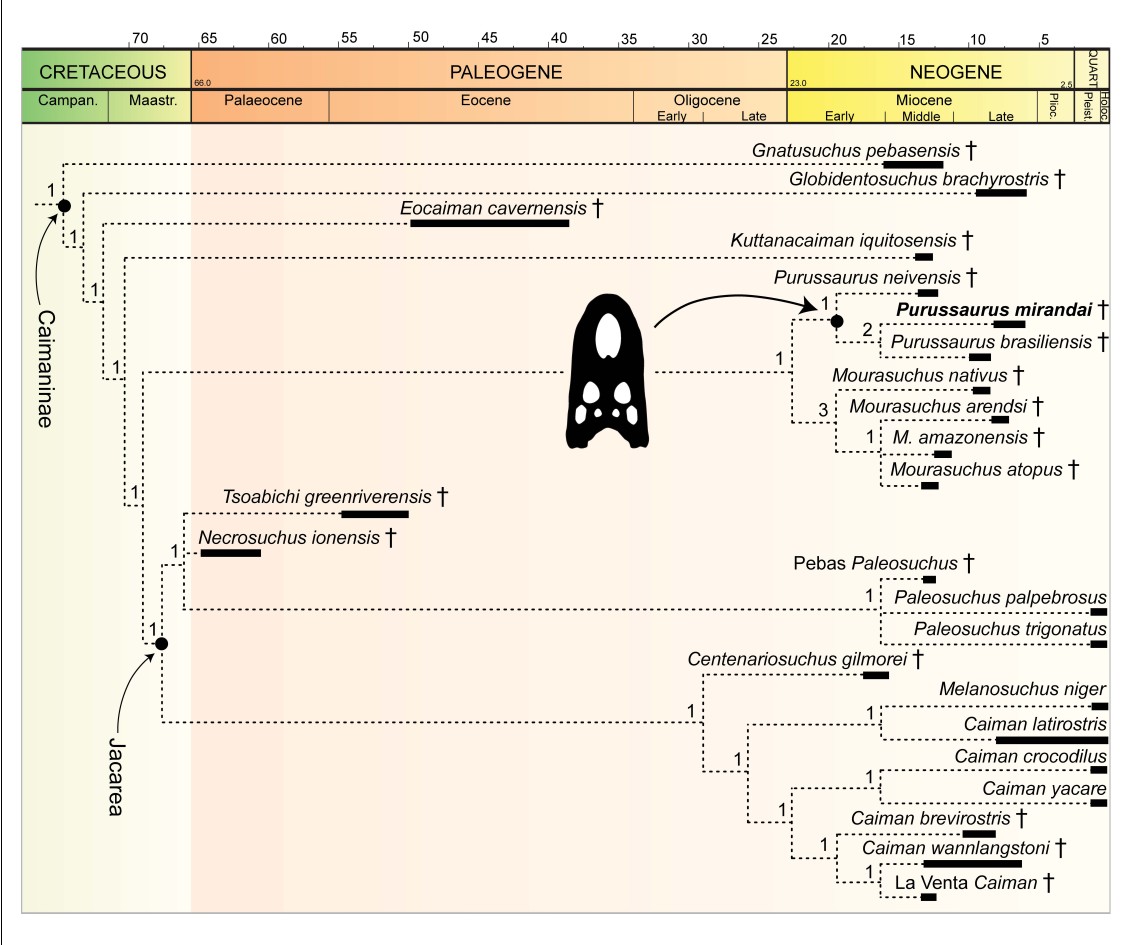

**Figure 1.** Phylogenetic analysis with updated scoring of *Purussaurus mirandai* based on AMU-CURS-541 (see Materials and methods for explanations). Here only an excerpt of the Caimaninae clade of the strict consensus tree is shown to serve as phylogenetic framework of *P. mirandai* (in bold). Bremer support values are given above the branches. For the remainder of the topology see the results section and *Figure 1—figure supplement 1* (see also *Salas-Gismondi et al., 2015*: supplementary fig. S6).

The online version of this article includes the following figure supplement(s) for figure 1:

**Figure supplement 1.** Full topology of the strict consensus tree of 20 most parsimonious trees (tree length = 687 steps; CI = 0.383, RI = 0.806) recovered in main analysis.

## Results

### Systematic palaeontology

Crocodylia Gmelin, 1789

 Alligatoridae Gray, 1844

 Caimaninae Brochu, 2003 (following *Norell, 1988*)

 *PURUSSAURUS* Barbosa-Rodrigues, 1892

 *P. mirandai* Aguilera, Riff and Bocquentin-Villanueva, 2006

 Holotype material: UNEFM-CIAAP-1369, comprising the skull and associated lower jaw material, as well as a femur and ischium, which, according to *Aguilera et al. (2006)* was collected at El Hatillo locality (see *Scheyer and Delfino, 2016* for locality information).

 New referred material: AMU-CURS-541, an associated but disarticulated skeleton, preserving cranial material and much of the postcranium, which was embedded normally with its abdomen in the sediment. The specimen was preliminarily assigned to *Purussaurus* cf. *P. mirandai* (*Scheyer and Delfino, 2016*), but given the overall shape and proportions of the lower jaw (see below), the lightly wavy lateral outline of the dentary, the low premaxilla and low and slender jugal (indicating a rather

flat skull profile) it is herein referred to as *Purussaurus mirandai*. The shape of the mandible, the teeth, and of the tooth row in general are otherwise also congruent with that of the holotype specimen of *P. mirandai*. There is so far no indication for the presence of a second species of *Purussaurus* in the Urumaco Formation. The phylogenetic context of the species among Caimaninae is shown in *Figure 1*.

Locality and age: The specimen comes from sediments of the Upper Member of the Urumaco Formation (late Miocene), from 'Norte El Picache' locality (11° 15′ 09.00′′ N; 70° 13′ 40.00′′ W), Urumaco, Falcón state, Venezuela.

Its most novel feature found in the sacral region (juncture of vertebral column and pelvis) expands the morphological diversity of the axial skeleton in crown crocodylians and suggests developmental changes correlated with biomechanical demands for support and locomotion, likely correlated with giant body size. Furthermore, we infer that the gigantic size of up to 10–12 meters of total body length and related mass of these animals (see also *Aureliano et al., 2015*) influenced the peculiar pectoral and pelvic morphology.

In the following, we thus provide an abbreviated description of the girdle bones and the sacral region of *P. mirandai* pertinent to the discussion (*Figure 2*, *Figure 2—figure supplement 1*, *Figure 3*; *Figure 3—figure supplement 1*). An exhaustive description of all studied specimens, including the postcranial bones of AMU-CURS-541, will be published elsewhere.

## Pectoral girdle

Of the pectoral girdle elements (*Figure 2*), isolated scapulae are known from two specimens (AMU-CURS-541; UNEFM-CIAAP-1367) and a well-preserved and isolated coracoid is known from UNEFM-CIAAP-1367. Similar to extant alligatorids such as *Alligator mississippiensis* and *Caiman crocodilus*, *Purussaurus mirandai* has narrow scapular blades. In contrast to the extant taxa, the extinct species has the scapulae oriented more dorsally and somewhat posteriorly, as well as ventromedially and slightly posteriorly oriented coracoids. The long, flat and narrow blade, which is set off by a constriction from the proximal articulation and the glenoid fossa, extends only slightly towards its distal end. The deltoid crest (acromion of *Cong, 1998*) of the scapula of *Purussaurus* from the Urumaco Formation (well preserved in UNEFM-CIAAP-1367) is a very robust and distinct process, and of similar proximodistal expansion as the width of the scapular articulation with the coracoid. The coracoid has a prominent and robust shaft, a narrow articular facet for the scapula, an ovoid coracoid foramen close to the articular facet, and an expanded, flaring distal part.

## Pelvic girdle

In general, these elements (*Figure 3*) resemble those of other crocodylians in overall shape, but show also some specializations. As in *A. mississippiensis* (e.g., *Brochu, 1999*), the ilium of *P. mirandai* (AMU-CURS-541, UNEFM-CIAAP-1369) has a weak indentation at the dorsal margin of the post-acetabular process. Most strikingly, however, the medial surface of the *Purussaurus* ilium shows three rugose concavities separated by ridges forming a 'π' (*Figure 3C–E*), instead of a 'τ' as in extant crocodylians with two sacral vertebrae. These concavities are the attachment sites for the sacral ribs and lie close together, their size responding to the anteroposterior width of each of the three sacral ribs as is visible in UNEFM-CIAPP 1369. All of the four recovered ilia assignable to *Purussaurus* from the Urumaco Formation show the three articulation sites separated by low bony ridges.

To date, there is only limited comparative information on the pelvic girdle morphology of other giant crocodylian forms. The well-preserved right ilium of UCMP 38012 (holotype) of *Mourasuchus atopus* from La Venta, Colombia, however, with 137 mm length (*Langston, 1965*); much smaller than *Purussaurus*; also has 'π'-shaped ridges on its medial side. Given the lack of other specimens corroborating the potential for three sacrals, we conservatively treat the condition in *Mourasuchus* as being inconclusive.

In some mekosuchine crocodylians (i.e., morphotype described as 'pelvic forms three and four'; see also *Figure 3—figure supplement 2*), the 'τ'-shape separation is not obvious, because the attachment sites of the two sacral ribs are shifting apart, thus creating a space in between (*Stein et al., 2017*). This space, however, consists of a smooth and flat bone surface, which should not be confused with an additional rugose and excavated attachment site.

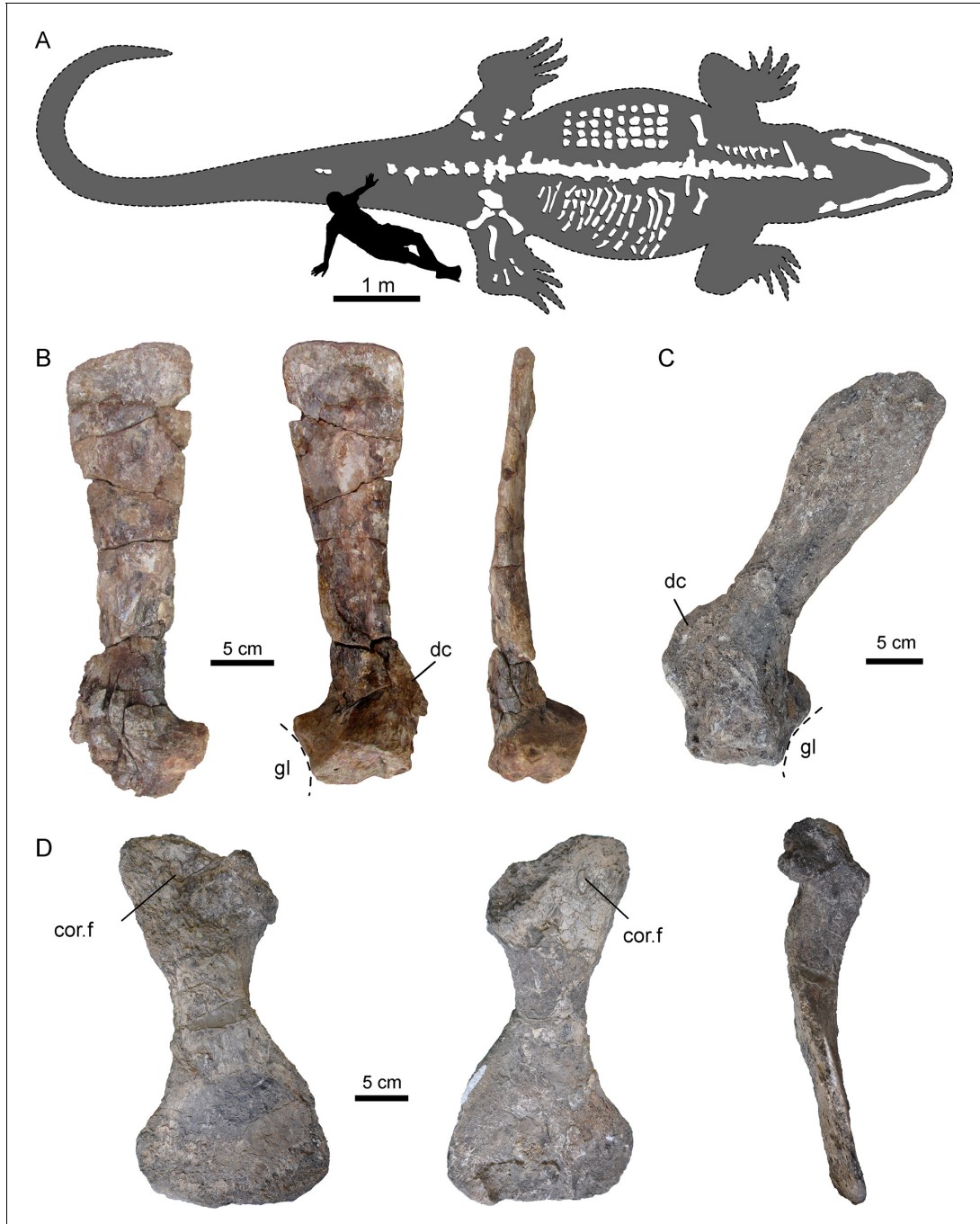

**Figure 2.** Selected pectoral bones of AMU-CURS-541 and UNEFM-CIAAP-1367 of *Purussaurus* from the Urumaco Formation of Venezuela. (A) Interpretative reconstruction of the complete body outline of *P. mirandai* (AMU-CURS-541) showing the preserved and assembled postcranial bones and the lower jaw in tentative live position. Osteoderms (in upper part of trunk) and ribs (in lower part of trunk) are not in life position. The second author (OS) serves as scale (see *Figure 2—figure supplement 1*). (B) Left scapula of AMU-CURS-541 in lateral, medial, and posterior view. (C) Right scapula of *Purussaurus* cf. *P. mirandai* (UNEFM-CIAAP-1367) in medial view. (D) Right coracoid (UNEFM-CIAAP-1367) in dorsomedial, ventrolateral, and anterior view. Abbreviations: cor.f, coracoid foramen; dc, deltoid crest of scapula; gl, glenoid fossa.

The online version of this article includes the following figure supplement(s) for figure 2:

**Figure supplement 1.** Specimen AMU-CURS-541 of *Purussaurus mirandai* in the field at 'North of El Picache' locality, Urumaco, Falcón state, Venezuela.

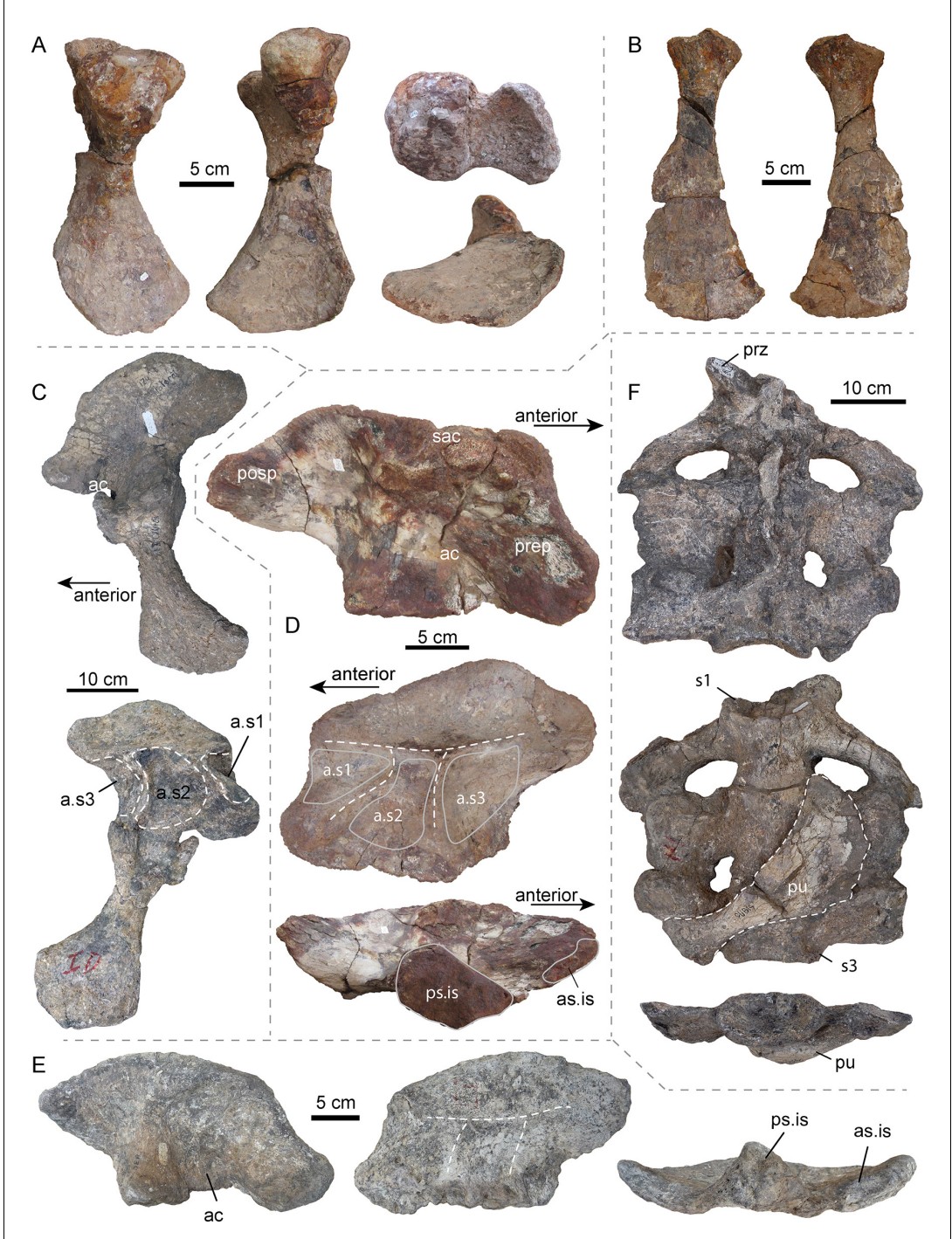

**Figure 3.** Selected pelvic and sacral bones of AMU-CURS-541 and other specimens of *Purussaurus* from the Urumaco Formation of Venezuela. ( **A**) Left ischium (AMU-CURS-541) in posterior, anterior, dorsal (above), and ventral (below) view. (**B**) Right pubis (AMU-CURS-541) in ventral and dorsal view. (**C**) Articulated left ilium and ischium (non-holotype, additional bones accessioned under UNEFM-CIAAP-1369) in lateral and medial view. Note the three large depressions separated by ridges, that is attachment sites for three sacral ribs (a.s1-a.s3). (**D**) Right ilium (AMU-CURS-541) in lateral, medial, and ventral view. (**E**) Right ilium (UNEFM-CIAAP-1367) in lateral, medial, and ventral view. Stippled white lines indicate the weathered ridges separating the articulation facets for the sacral ribs. (**F**) Sacral region (non-holotype, additional bones accessioned under UNEFM-CIAAP-1369) consisting of three sacrals in articulation in dorsal, ventral, and posterior view. Note the right pubis (in dorsal view) attached ventrally to the sacrals. Abbreviations: ac, acetabulum; a.s1-a.s3, attachment sites for three sacral ribs; as.is, anterior articular surface for ilium; posp, postacetabular process; prep, preacetabular process; prz, prezygapophysis; ps.is, posterior articular surface for ischium; pu, pubis; s1-s3: sacral vertebrae/ribs 1–3; sac, supraacetabular crest. The online version of this article includes the following figure supplement(s) for figure 3:

*Figure 3 continued on next page*

*Figure 3 continued*

**Figure supplement 1.** Selected sacral vertebrae and long bones of *Purussaurus* from the late Miocene Urumaco Formation.
**Figure supplement 2.** Right ilium (UCMP 38012) of *Mourasuchus atopus*, La Venta, Colombia [picture courtesy: P. Holroyd, Berkeley] and left ilia of mekosuchine 'pelvic forms three and four' (from *Stein et al., 2017*, figures 6B and 7B; images published under Creative Commons CC-BY 4.0).

The pubis is hatchet-shaped in *P. mirandai,* with a slender shaft and distal flaring blade. The proximal bone surface of the pubis differs from that of other crocodylians (*Claessens and Vickaryous, 2012*; *Mook, 1921b*; *Stein et al., 2017*) in that the lateral part of the surface is angled.

The ischium is known in the holotype UNEFM-CIAAP-1369 and non-holotype material of *P. mirandai*, as well as in AMU-CURS-528 and AMU-CURS-541. The ischium has a slender shaft, and proximally two articulation facets for the ilium, which form the acetabulum, and a distally flaring broad shield. Medially the ischia have a straight contacting margin.

## Sacral region

Isolated sacrals are known in specimens AMU-CURS-541 (*Figure 3—figure supplement 1*) and MCNC-URU-111–72V, but it is the complete articulated sacrum in UNEFM-CIAAP-1369 (non-holotype material), consisting of three moderately preserved but articulated sacral vertebrae that is the most informative. All three sacral vertebrae have the centrum and sacral ribs preserved in articulation, showing the detailed morphology of the pelvic girdle and the connection of the ilium with the three sacral vertebrae in *Purussaurus*. The sacral ribs of the first sacral (=dorsosacral) vertebra are tilted posteriorly. The sacral ribs of the second sacral extend laterally and the sacral ribs of the third sacral are tilted anteriorly. Of the three sacrals, the ribs of the mid-sacral show the largest anteroposterior expansion, followed by the third sacral and then those of the first sacral, which are the least expanded. With the exception of a single prezygapophysis, the other zygapophyses and the neural spines of the sacrals are missing or strongly distorted. Of the isolated sacrals, the 3$^{rd}$ sacral (=primordial sacral 2) centrum of AMU-CURS-541 shows the best-preserved articular surfaces, indicating that the anterior condyle is round in shape and flattened (rather than strongly convex), whereas the posterior cotyle is slightly oval-shaped and concave. The cotyle also shows a thick marginal rim, but a flange seems to be absent (*Figure 3—figure supplement 1D,E and G*). All above-mentioned specimens are considered large adults, and they vary only modestly in shape. The posterior articular surface of the last vertebral centra in the sacral series (i.e., primordial sacral 2) of the non-holotype material accessioned under UNEFM-CIAAP-1369 is 12 cm wide and 6 cm tall (distance between sacral rib ends is 30 cm), whereas that of primordial sacral 2 of AMU-CURS-541 is about 12.8 cm wide and 8.8 cm tall (distance between sacral rib ends is 28 cm, but the ends are not complete). The strongly weathered posterior articular surface of the sacral rib of MCNC-URU-111–72V is 9 cm wide and 5.5 cm tall (sacral rib ends are 36 cm apart). Given the strongly concave posterior margins and posterior tilting of its sacral ribs, and in comparison with the complete sacral series of UNEFM-CIAAP-1369 described above, this specimen is identified as an isolated dorsosacral, that is the first sacral in the series, as well (*Figure 3—figure supplement 1J–L*).

## Phylogenetic analyses

The osteological description and analysis provided 43 characters that could be scored based on AMU-CURS-541, with 19 (15 postcranial, four mandibular) new in comparison to the previously available scoring of *P. mirandai* (*Salas-Gismondi et al., 2015*).

## Phylogenetic analysis with AMU-CURS-541 merged with *P. mirandai* scores

The TNT analysis recovered 20 most parsimonious trees (tree length = 687 steps; CI = 0.383, RI = 0.806), for which a strict consensus tree was computed. The analysis (*Figure 1*) shows the best resolution of those performed herein. It yielded a similar topology to that presented by *Salas-Gismondi et al. (2015)*: fig. 4) with the following exceptions: Within Gavialoidea, the 'Pebas gavialoid' *Gryposuchus pachakamue* (*Salas-Gismondi et al., 2016*), fell into a polytomy with *Siquisiquesuchus, Ikanogavialis, Piscogavialis, Gavialis,* and a *Gryposuchus* clade (consisting of *G. croizati* and *G. colombianus*). The *Purussaurus* clade – sister to the genus *Mourasuchus* - is fully resolved with *P.*

*neivensis* being the sister to the clade *P. mirandai* – *P. brasiliensis*. The Jacarea clade is better resolved compared to the original analysis of *Salas-Gismondi et al. (2015)*, but Bremer support for the clades within Caimaninae was also generally low (*Figure 1*; *Figure 1—figure supplement 1*).

### Additional analysis with AMU-CURS-541 added as separate terminal taxon

This analysis recovered 60 most parsimonious trees (tree length = 687 steps; consistency index CI = 0.383, retention index RI = 0.806), for which a strict consensus tree was computed. The analysis shows a similar topology as presented in the main text (see also *Salas-Gismondi et al., 2015*, fig. 4). The sister grouping of *P. mirandai* and *P. brasiliensis* could not be recovered and instead the *Purussaurus* clade collapsed into a polytomy.

### Additional analysis with AMU-CURS-541 merged with *P. mirandai* scores and *Melanosuchus fisheri* included

This analysis yielded 160 most parsimonious trees (tree length = 688 steps; CI = 0.382, RI = 0.805). The strict consensus of those trees showed the same topology as in a previous analysis of *Salas-Gismondi et al. (2015)*, with the exception of the polytomy with *Siquisiquesuchus*, *Ikanogavialis*, *Piscogavialis*, *Gavialis*, a *Gryposuchus* clade within Gavialoidea, and a poorly resolved Jacarea clade.

## Discussion

With the description of AMU-CURS-541 (including ca. 100 preserved elements) and other specimens from the Urumaco Formation, novel data on the postcranial anatomy of the giant caimanine *Purussaurus mirandai* are now available. Our phylogenetic analyses enabled by these data underscore the monophyly of *Purussaurus*, as sister taxon to the *Mourasuchus* clade, and both groups as sister clade to crown-group caimans (*Figure 1*) within Caimaninae. The sister group relationship of *P. mirandai* and *P. brasiliensis* persists after the revised scoring of *P. mirandai* with AMU-CURS-541. The better resolution of the Jacarea clade also indicates that the questionable scoring of *Melanosuchus fisheri* is likely responsible for the polytomy encountered by *Salas-Gismondi et al. (2015)*. The studies of *Bona et al., 2018* and *Souza-Filho et al., 2019* encountered similar problems, with the former – excluding *Me. fisheri* – having a better resolved Jacarea clade, and the latter – including *Me. fisheri* – showing less overall resolution, especially in the Jacarea clade. Future analyses should use the updated scoring of *P. mirandai* and a critical revision of the scorings of *Melanosuchus fisheri* and *Melanosuchus niger* is recommended.

### Pectoral region

The scapula and coracoid remained separate in *Purussaurus*, even in large specimens (*Figures 2* and *4*A), whereas in extant caimans, scapulocoracoid synchondrosis closure begins relatively early in ontogeny. This early closure was suggested as a peramorphic feature of Caimaninae (*Brochu, 1995*), as well as an ambiguous character support of the group (*Brochu, 1999*). The scapula blade is oriented only slightly posteriorly, and the coracoid is ventromedially and slightly posteriorly oriented, giving the whole pectoral girdle a rather upright or straight (subvertical) appearance in lateral view (*Figure 4A*). The scapulocoracoid suture in the *Purussaurus* species is narrow in comparison to that in extant species because both pectoral girdle bones lack a proximal anterior expansion (*Figure 4A*). The very wide and robust deltoid crest of the scapula of *Purussaurus* indicates a well-developed origination site of the *deltoideus clavicularis* muscle (*Meers, 2003*; = M. *deltoideus scapularis inferior* sensu *Brochu, 1999*). The *deltoideus clavicularis* likely strengthened the anchoring of the humerus to the shoulder girdle. Analogous expansions of the insertion of this muscle onto the proximal portion of the humerus in the baurusuchid crocodyliform *Stratiotosuchus* were cited as evidence for improved parasagittal limb function (*Riff and Kellner, 2011*). The deltoid crest may also have hosted parts of the insertions of the *levator scapularis* and *trapezius/cucullaris* muscles (*Cong, 1998*), which should have had functions related to stabilization/mobilization of the pectoral girdle or the neck.

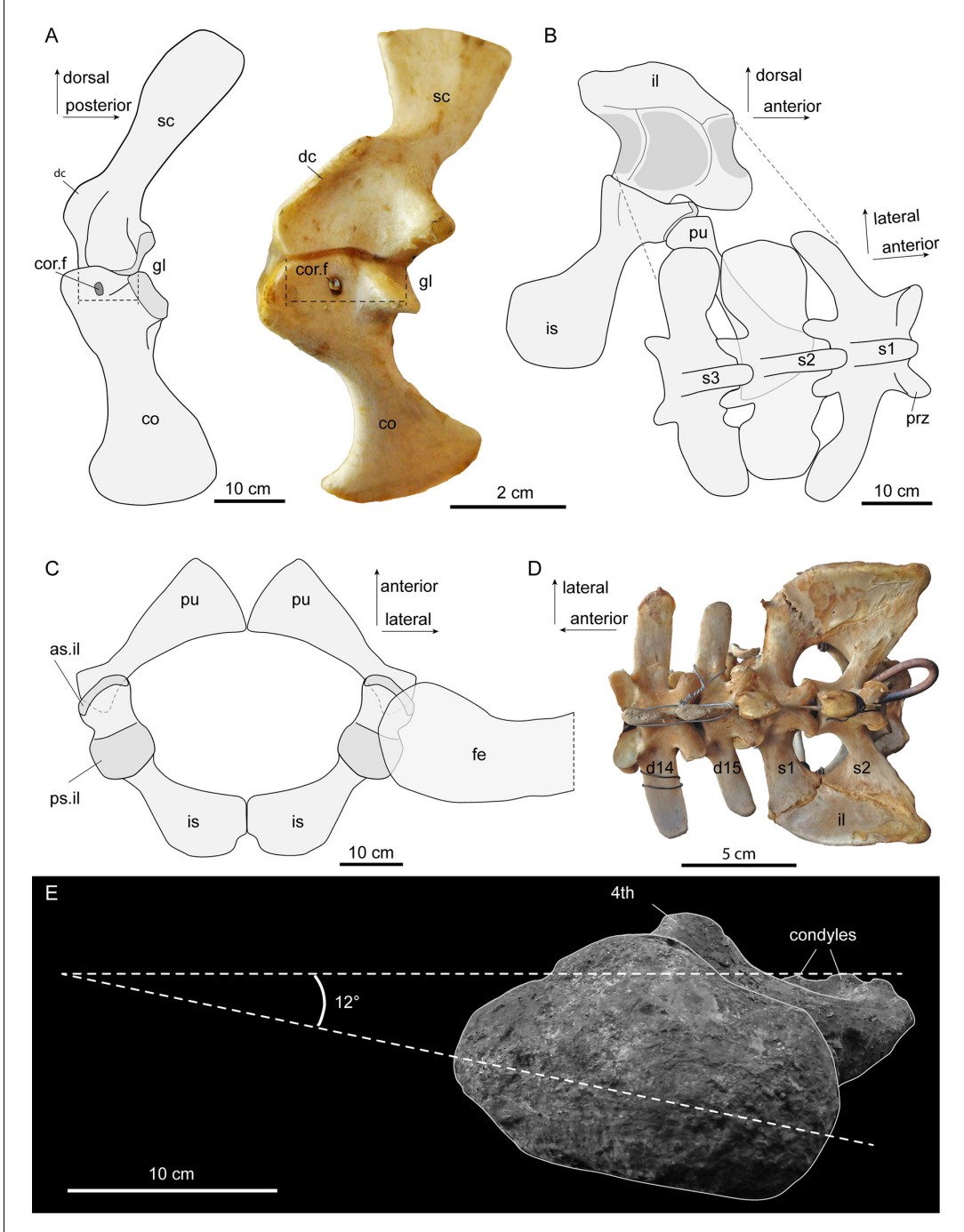

**Figure 4.** Interpretative sketches of girdle articulation in *Purussaurus*, based on the girdle bones of several specimens (AMU-CURS-541; UNEFM-CIAAP 1367; UNEFM-CIAAP-1369) in comparison to selected extant bones. (A) Right pectoral girdle in medial view in comparison with a left (mirrored) pectoral girdle of an extant *Crocodylus niloticus*. Stippled lines indicate the width of the bony articulation between scapula and coracoid. (B) Left pelvic girdle in medial view and sacral vertebral portion in angled dorsal view. (C) Pubes and ischia (bones were mirrored for completion) and superimposed femoral head in dorsal view. (D) Dorsosacral transition of the vertebral column in an extant *Caiman yacare*. (E) Proximal view of femur revealing low torsion (12°) of the femoral head to the plane of the distal condyles. Abbreviations: as.il, anterior articular surface for ilium; 4th, fourth trochanter; co, coracoid; cor.f, coracoid foramen; d14-d15, the 14th and 15th dorsal vertebra; dc, deltoid crest; fe, femur; gl, glenoid fossa; il, ilium; is, ischium; pu, pubis; prz, prezygapophysis; ps.il, posterior articular surface for ilium; s1-s3, sacral vertebrae and ribs 1–3; sc, scapula.

## Pelvic and sacral region

The pelvic girdle bones of *Purussaurus* articulate in similar fashion to those of extant crocodylians (*Figure 4B,C*; see *Claessens and Vickaryous, 2012*), with a double articulation between ilium and ischium and a sole articulation of the pubis with the anterior margin of the ischium. The angle of the proximal articular facet in the pubis in *P. mirandai* is peculiar, because the standard articular surface in crocodylian taxa has a sub-circular outline and a single, weakly concavoconvex articular surface with the ischium (e.g., *Claessens and Vickaryous, 2012*). The lateral, angled part of the proximal articular facet in the *Purussaurus* pubis is thus impossible to articulate with the ischium when the medial part is in articulation (*Figure 4C*). The medial attachment areas between the left and right distal pubes were likely a point contact (*Figure 4C*); compared to extant crocodylians, the medial contact between the ischia was reduced in length.

In addition to isolated sacrals, the complete sacral region with the three sacrals of UNEFM-CIAAP-1369 is known for *P. mirandai*. Together with the evidence from the medial articulation sites on the ilia, there is strong support for the 'three sacral' condition in this extinct caimanine as a unique trait within crown-Crocodylia (*Böhmer et al., 2015*; *Mook, 1921b*; *Romer, 1956*). Based on comparison with extant crocodylian skeletons (e.g., *Crocodylus niloticus*, *Caiman yacare*; *Figure 4D*) and the shapes of the sacral ribs and associated attachment sites on the ilia, we show that the last dorsal (i.e., lumbar) vertebra has been sacralised into a dorsosacral in *P. mirandai*. This leads to a vertebral count of nine cervicals, 14 dorsals, and three sacrals. The first caudal in *P. mirandai* has, as in extant crocodylians, a biconvex centrum shape, with the transverse processes being posteriorly oriented.

The femoral head of the best-preserved *P. mirandai* femur appears to have a more medially directed orientation than in most other crown-Crocodylia, at only ~12° relative to the mediolateral axis of the femoral condyles (*Figure 4E*). While bone distortion and wear of the condyles might have altered this orientation, the low torsion of the femoral head compared to condyles contrasts even with the relatively medial 36° orientation in *Stratiotosuchus* or the larger angles found in extant crocodylians (*Riff and Kellner, 2011* reported an angle of 52° in a specimen of *Caiman yacare*).

## Congenital vertebral anomalies

In humans, the pathological condition of a congenital malformation lumbosacral transitional vertebra (LSTV) is widespread and frequently studied in the medical and veterinary medical literature. This condition is proposed to be linked to lower back pain symptoms referred to as Bertolotti's syndrome in humans (see *Holm et al., 2017*; *Jancuska et al., 2015*) for overviews) or predispose for cauda equina syndrome in dogs (e.g., *Flückiger et al., 2006*). The developmental mechanisms underlying these pathologies occur early during embryogenesis, when *Homeobox* gene expression, that is *Hox8*, *Hox10*, and *Hox11* (the latter two also playing an important role in limb patterning), induces axial patterning of the posterior dorsal (=lumbar), sacral, and anterior caudal regions in the embryo (*Casaca et al., 2014*; *Swinehart et al., 2013*; *Wellik, 2007*; *Wellik and Capecchi, 2003*). *Hoxc8* is expressed throughout the dorsal (=thoracolumbar) region of the crocodylian skeleton, but not in the sacral region (*Böhmer et al., 2015*; *Böhmer, 2013*; *Mansfield and Abzhanov, 2010*). Slight heterochronic shifts in the interplay of *Hox* gene expressions are known to cause drastic shifts in development (*Gérard et al., 1997*), as in *Hoxa10* and *Hoxa11* expression in the sacral region (*Carapuço et al., 2005*). The latter study showed that changes in the timing of expression of *Hoxa10* and *Hoxa11* within the presomitic mesoderm versus expression in the somites influenced rib development, sacralisation, and caudal formation in the sacral region. As such, *Hoxa10* expression in the presomitic mesoderm led to formation of vertebrae without ribs, whereas the same expression in the somites led to vertebrae with ribs; similarly, *Hoxa11* expression in the presomitic mesoderm caused sacralisation, whereas in the somites it induced caudal formation.

Among extant crocodylians, numerous pathological conditions affecting the axial skeleton have been described (*Kälin, 1933*; *Rothschild et al., 2012*). These reportedly include changes in the number and identity of vertebrae, such as vertebral addition in the dorsal series, shifting of the sacral region, sacralisation of last dorsals and first caudals (*Baur, 1886*; *Baur, 1889*; *Reinhardt, 1873*; *Reinhardt, 1874*), but images or drawings were usually not provided, and only occasionally the studied specimens were directly referred to (e.g., *Baur, 1889*), p. 240). A unilateral articulation of a left transverse process of the last, hemisacralised dorsal vertebra with the left ilium has also been

recently noted and figured in a pathological specimen of *Alligator mississippiensis*, while discussing the repeated independent appearance of non-pathological dorsosacrals among Triassic Archosauriformes (*Griffin et al., 2017*). The latter study discussed the potential role of *Hox* genes, especially *Hox11* paralogs, and changes in the timing of expression thereof, for compartmentalizing the dorsal and sacral series and shifting the boundary between them in the phytosaur *Smilosuchus* from the Upper Triassic Chinle Formation of northeastern Arizona, USA.

In our sample of comparative materials, we discovered a hemisacralised last dorsalsacral in a juvenile specimen of the dwarf caiman *Palaeosuchus palpebrosus* (*Figure 5*; specimen RVC-JRH-PP4). This specimen exhibits the girdle bones and stylopodial elements in life position/articulation and thus serves well for comparison to the *Purussaurus* fossils. It has a wide scapulocoracoid synchondrosis, and a more strongly torsioned femoral head (*Figure 5C*; with 53.5˚ similar to angle reported for *Caiman* in *Riff and Kellner, 2011*) compared to the distal condylar plane (*Figure 5C,D*). As reported for the *A. mississippiensis* specimen (*Griffin et al., 2017*) mentioned above, the primordial sacral 1 of *P. palpebrosus* (*Figure 5D,E*) shows the development of a shallow flange onto the dorsosacral and a slight posterior shift of its sacral rib base on the side where the hemisacralization occurred.

In addition, another pathological specimen of *P. palpebrosus* (MACV-6139; image courtesy of Mariano Padilla Cano) shows a pathological 'three-sacral' condition in which the size and shape of the sacral ribs and the articulation sites on the medial surfaces of the ilia varies between the left and right side (*Figure 5—figure supplement 1*).

To our knowledge, *Purussaurus mirandai* is the first member of crown-group Crocodylia with three non-pathological sacrals (see *Müller et al., 2010*). A similar condition was reported in the terrestrial notosuchian crocodylomorph *Notosuchus* from the Late Cretaceous of Argentina, which has the second and third sacral fused, in some atoposaurid neosuchians with three unfused sacrals (*Fiorelli and Calvo, 2008*; *Nobre and Carvalho, 2013*; *Pol, 2005*; *Tennant and Mannion, 2014*), and in some teleosauroid Machimosaurini, a group of large-sized marine crocodylomorphs from the Jurassic, in which the first caudal was sacralised (e.g., *Johnson et al., 2018*; *Jouve et al., 2016*). All of these groups, however, differ strongly from the conserved body plan of extant crocodylians.

## Morphofunctional interpretations

*Molnar et al. (2014)* identified the lumbosacral joint (between dorsal 15 and sacral 1) of the vertebral column of *Crocodylus niloticus* as having the highest intervertebral joint stiffness, which had been hypothesized as beneficial in supporting a large tail and countering hind limb forces (*Willey et al., 2004*). Based on a study of five small specimens (between 40 and 50 cm snout-vent length) of *Alligator mississippiensis*, *Willey et al. (2004)* found the centre of mass to be situated about 70% along the trunk length, and the tail mass to be up to 28% of the total body mass. In addition, *Aureliano et al. (2015)* estimated the largest specimens of *Purussaurus brasiliensis* to have reached more than 12 m in body length and over eight metric tons in mass.

Given the 145 cm length of the preserved lower jaw and the size of the vertebrae with up to 11.8 cm centrum length, AMU-CURS-541 is estimated to have ranged between eight to ten meters in total length. Unfortunately, the associated femur of AMU-CURS-541 was only partially preserved, making size estimations based on this skeletal element difficult (*Farlow et al., 2005*). Comparisons to other specimens of *Purussaurus* from the Urumaco Formation indicate, however, that AMU-CURS-541 is not the largest individual recovered so far. In AMU-CURS-20, the largest preserved vertebra (13.6 cm centrum length) is about 15% larger compared to the vertebrae of AMU-CURS-541. In addition, the dimensions of the associated and fairly complete right femur of AMU-CURS-20 (54 cm length; compared to 51 cm of the femur of the holotype UNEFM-CIAAP-1369 of *P. mirandai*) are among the largest crocodylian femora known (*Salas-Gismondi et al., 2007* reported on a 54.5 cm long femur of *Purussaurus* from Peru). With 210 mm, the femur of AMU-CURS-20 has a similar minimum shaft circumference as the femoral shaft fragment MNN G102–2 of *Sarcosuchus imperator*, which was used by *Farlow et al. (2005)* to estimate the total length of that specimen to range between 7.2 and 9.1 m.

We used the dataset and R software code from *O'Brien et al. (2019)* with *P. mirandai* input into the phylogeny as per *Figure 1* and with a branch length of 0.01 (assuming 25% corrected mass as per their methods) to estimate the total body mass and total length of specimen AMU-CURS-541 (head width ~0.88 m). Based on this more conservative approach, our analysis estimated 1686–2637 kg and 7.11–8.01 m for total body mass and length, respectively (from lower to upper interquartile

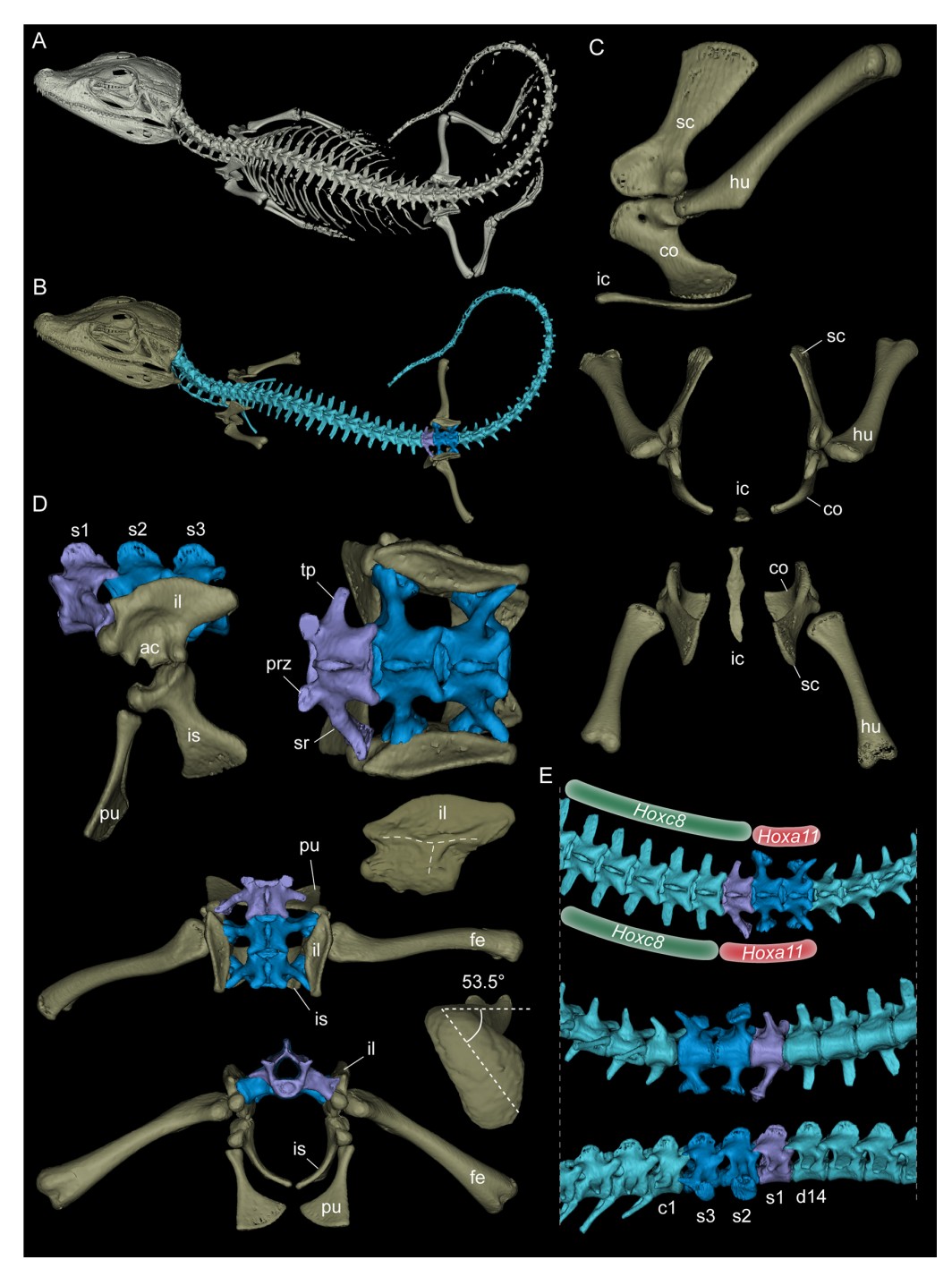

**Figure 5.** Rendered 3D-models of a juvenile dwarf caiman *Palaeosuchus palpebrosus* (RVC-JRH-PP4; not to scale). (**A**) Rendering of the complete specimen. (**B**) Model with only the axial and partial appendicular skeleton (in dorsal view) highlighted. Vertebrae are shown in light blue, except those of the sacral region, in which the dorsosacral in lavender and the true or primordial sacrals in dark blue. (**C**) Pectoral girdle elements in left lateral, anterior, and dorsal view. (**D**) Sacral region in left lateral, dorsal (and dorsal with femur included), and anterior view. Note asymmetry of the dorsosacral. The medial view of the right ilium reveals the 'τ' ridges separating the two articular surfaces of the sacral ribs. The medial side of the left ilium (not shown) looks similar, because the sacralised rib of the dorsosacral articulated only slightly with the anterior margin of the ilium, leaving no deep articular scar. In the right femur, a torsional angle of 53.5° of the head to the plane of the distal condyles was measured. (**E**) Focus on the posterior dorsal, sacral and anterior caudal series of the skeleton in dorsal, ventral and right lateral view. The inferred asymmetrical shift of the domain boundaries of *Hoxc8* and *Hoxa11* leading to congenital malformation lumbosacral transitional vertebra in this specimen of *P. palpebrosus* is indicated.

*Figure 5 continued on next page*

*Figure 5 continued*

Abbreviations: ac, acetabulum; c1, first caudal vertebra; co, coracoid; d14, 14th dorsal vertebra; fe, femur; hu, humerus; ic, interclavicle; il, ilium; is, ischium; prz, prezygapophysis; pu, pubis; s1-s3: sacral vertebrae/ribs 1–3; sc, scapula; sr, sacral rib; tp, transverse process.

The online version of this article includes the following figure supplement(s) for figure 5:

**Figure supplement 1.** Close-up of the sacral region of specimen MACV-6139 of the extant dwarf caiman *Palaeosuchus palpebrosus* [picture courtesy: M. Padilla Cano, Madrid].

ranges of Bayesian analysis). This compares favorably with other large Crocodyliformes such as *Sarcosuchus imperator* (as per *O'Brien et al., 2019* 'longirostrine' calculations), estimated at 1976–2981 kg and 8.50–9.47 m (their Table 5). Compared to the more massive and robust appearance of *P. brasiliensis*, the more 'gracile' *P. mirandai* specimen AMU-CURS-541 could have had the tail alone weighing up to 472–738 kg (based on *Willey et al., 2004*; values that exceed most average masses of extant crocodylians (*Grigg and Kirschner, 2015*); the exceptionally large 'Lolong', an old *Crocodylus porosus* 6.17 m long, was reported to be 1075 kg: *Britton et al., 2012*).

Regardless of the mass estimation used, we speculate that the 'three sacrals' condition, together with the robustness of the hind limb bones encountered in *P. mirandai,* was related to these animals' giant size and body mass (*Aureliano et al., 2015*; *Salas-Gismondi et al., 2007*). A longer sacrum with three instead of only two sacrals articulating with the ilium expands the connection between the axial skeleton and the posterior appendicular skeleton, thus potentially increasing stability of this region in the vertebral column and allowing for better transmission of forces across the pelvic and caudal regions.

Comparative data from other giant crocodylians, however, are either scarce or remain largely undescribed (e.g., *Deinosuchus*: *Colbert and Bird, 1954*; *Holland, 1909*; *Rivera-Sylva et al., 2011*; *Schwimmer, 2002*; *Euthecodon*: *Storrs, 2003*; *Gryposuchus*: *Langston, 1965*; *Riff and Aguilera, 2008*; *Laganosuchus*: *Sereno and Larsson, 2009*; *Mourasuchus*: *Langston, 1965*; *Sarcosuchus*: *Dridi, 2018*; *Sereno et al., 2001*). Thus, the influence of body size and mass on the sacral region of giant crocodylians remains largely unexplored. Of those taxa, at least one well-preserved ilium from *Mourasuchus atopus* shows a 'π' pattern of ridges on its medial surface (UCMP 38012, *Figure 3— figure supplement 2*). This could potentially indicate the presence of three sacrals in this additional giant caimanine if confirmed by further specimens. The 'three-sacral' condition described herein is purely of functional origin, as both *Mourasuchus* and *Purussaurus* are giants (although likely of distinctly different body masses), or there may also be a phylogenetic signal involved, as both taxa are sister genera within Caimaninae.

Other studies of Crocodylomorpha/Archosauria have inferred that expanded ilia and sacralisation correlate with more upright limb posture and/or improved support against gravity (e.g., *Riff and Kellner, 2011*). The above evidence from the iliosacral region corroborates other specializations for more upright limb orientation or simply weight support in *P. mirandai*, including the more vertically oriented pectoral girdle and low torsion of the femoral head relative to the condyles.

## Conclusions

The pectoral and pelvic regions of *Purussaurus mirandai* show a peculiar morphology in comparison to extant taxa, including a narrow scapulocoracoid contact, wide and prominent deltoid crest on the scapula serving as a muscle origin, more vertically oriented pectoral girdle, and autapomorphic 'three sacral' condition in the pelvis, interpreted herein to be linked to the giant body size. We infer that the underlying developmental mechanisms, especially an earlier timing of expression of *Hox11* and partial suppression of *Hox10* (*Wellik and Capecchi, 2003*), resulting in shifted domains of *Hoxc8* and *Hox11*, led to the formation of the dorsosacral in *P. mirandai*. Discovery and examination of additional specimens of *Purussaurus* with exceptional preservation to document genus variation, and of other giant crocodylians would lead to a better understanding of the anatomical peculiarities of these taxa. Such studies should inspect the medial side of the ilium to provide indirect information on the sacral count.

## Materials and methods

Specimens of *Purussaurus* used in this study derive from different localities in the late Miocene Urumaco Formation, Falcón State, Venezuela. AMU-CURS-541, a specimen collected in 2013, consists of associated cranial, lower jaw and postcranial material. Our study also included: UNEFM-CIAAP-1367 (a largely undescribed specimen including pectoral and pelvic girdle elements), UNEFM-CIAAP-1369 (holotype and non-holotype referred material, the latter including a complete articulated sacrum), MCNC-URU-111–72V and MCNC-URU-158–72V (mixed assemblages of *Purussaurus*), AMU-CURS-020 (two femora associated with two vertebrae), and AMU-CURS-528 (postcranial material associated with disarticulated skull remains).

Extant comparative materials included specimens of *Crocodylus* and *Gavialis* in PIMUZ, Caimaninae from the collections in MACUT, as well as material stored and scanned by the RVC (see below for abbreviations).

### Phylogenetic analysis

The numerous postcranial elements of the new specimen AMU-CURS-541 were used to re-score some hitherto unknown characters for the species *Purussaurus mirandai*. All characters are taken from *Salas-Gismondi et al. (2015)*, a matrix largely based on and modified from *Brochu (2011)* and *Jouve et al. (2008)*.

Three maximum parsimony analyses were performed using TNT v. 1.5 (*Goloboff and Catalano, 2016*). In the first analysis, AMU-CURS-541 was scored as a separate terminal taxon besides the three species of *Purussaurus: P. mirandai*, *P. neivensis*, and *P. brasiliensis*. For the second analysis, the new scorings based on AMU-CURS-541 were added to the previous scorings of *P. mirandai*. Consistency indices and Bremer support values were calculated in TNT v. 1.5 using the 'stats.run' and 'Bremer.run' scripts (downloaded on 02.11.2018 from http://phylo.wikidot.com/tntwiki; see also http://gensoft.pasteur.fr/docs/TNT/1.5/) and the latter was checked manually by collapsing tree topologies through inclusion of successive suboptimal trees. In both phylogenetic analyses, the extinct taxon *Melanosuchus fisheri* was removed due to recently revealed inconsistencies between the holotype and referred material (*Bona et al., 2017*; *Foth et al., 2018*). In the third analysis, the updated scoring of *P. mirandai* was used, but *Me. fisheri* was left in, to see how the addition or the removal of that taxon influenced the results. In all analyses, the heuristic search (traditional search; space 60000 trees in memory, random seed = 1, Tree Bisection Reconnection (TBR) mode activated) was run with *Bernissartia fagesii* as outgroup, 1000 random additional sequence replicates and 100 trees saved per replication. Characters were equally weighted, unordered, and set as non-additive.

### Scores for phylogenetic analyses

Scorings of 201 characters used for *P. mirandai* updated by the scores of AMU-CURS-541 in the matrix of *Salas-Gismondi et al. (2015)*:

 ????1 110?? 00001 0?0?0 11110 01??? ??111 0?11? ?01?? ?1100
 00?2? ?1?11 ?1201 100?1 1001? ???0? 112?0 0021? 10000 11100
 ???10 00??? ?0000 00011 11?1? 11021 0?111 110?1 10200 10112
 000?? 10104 ????? ????? 0011? 2?00? 10??0 01000 00??1 00?10 0

Scorings of 201 characters used for AMU-CURS-541 as terminal taxon in the matrix of *Salas-Gismondi et al. (2015)*:

 ????1 110?? 00001 0?0?0 11110 0???? ??111 ??11? ?01?? ?1100
 00?2? ????1 ??2?? ?00?1 1001? ???0? ????? ????? ????? ?????
 ????? ????? ????? ????? ????? ????? ????? ????? ????? ?????
 ????? ????? ????? ????? ????? ????? ????? ????? ????? ???1? ?

### Institutional abbreviations. AMU-CURS,

Colección Paleontológica de la Alcaldía Bolivariana de Urumaco, Estado Falcón, Venezuela; MACUT, Collections of the Museo di Anatomia Comparata dell'Università di Torino hosted by the Museo Regionale di Scienze Naturali Torino, Italy; MACV, Museum of Comparative Anatomy of Vertebrates, Complutense University of Madrid, Spain. MCNC, Museo de Ciencias Naturales de Caracas, Venezuela; MNN, Musée National du Niger, Niamey, Niger; PIMUZ, Palaeontological Institute and Museum, University of Zurich, Switzerland; QM, Queensland Museum, Brisbane, Australia; RVC,

Royal Veterinary College, London, UK; UNEFM-CIAAP, Universidad Nacional Experimental Francisco de Miranda/Centro de Investigaciones Antropológicas, Arqueológicas y Paleontológicas, Coro, Venezuela.

## Acknowledgements

We thank the Alcadía Bolivariana de Urumaco and the Instituto del Patrimonio Cultural de Venezuela (IPC) for the authorization to collect and study the specimens. Hyram Moreno (MCNC), Gina Ojeda and Camilo Morón (UNEFM-CIAAP), and José Hernández (Museo Paleontológico de Urumaco) for their valuable support. Elena Gavetti and Franco Andreone (Museo Regionale di Scienze Naturali, Torino) are thanked for the loan of comparative osteological material. Austin Hendy (LACM), Maurice Joss (ZHDK), Daniel Núñez (PIMUZ), and Alex Hastings (VMNH) are acknowledged for fruitful discussions. Paula Bona (Univ. La Plata) and Rodolfo Salas-Gismondi (Universidad Peruana Cayetano Heredia, Lima) provided images and information on extant *Caiman* specimens. Pat Holroyd (Univ. of California at Berkeley) kindly provided images of *Mourasuchus atopus*, Carl Mehling (AMNH) images of *Deinosuchus*, Mariano Padilla Cano (MACV) images of *Palaeosuchus palpebrosus*, whereas Tyler Keillor (Univ. Chicago) is thanked for providing images of *Sarcosuchus imperator*. The program TNT is kindly made available through the sponsorship of the Willi Hennig Society. We thank Crocodiles of the World (Brize Norton, UK) for providing the scanned *Palaeosuchus* specimen. Finally, we thank reviewers Michelle Stocker and Chris Brochu, the editors John Long and Diethard Tautz, as well as Judith Recht for comments that led to manuscript improvement. This study was financially supported by the Swiss National Science Foundation (grant nos. 149506 and 162775 to TMS), and Generalitat de Catalunya (CERCA Program), and Spanish Agencia Estatal de Investigación (CGL2016-76431-P, AEI/FEDER, EU to MD). JRH was supported by funding from the European Research Council (ERC) under the European Union's Horizon 2020 research and innovation programme (grant agreement #695517).

## Additional information

### Funding

| Funder | Grant reference number | Author |
|---|---|---|
| Schweizerischer Nationalfonds zur Förderung der Wissenschaftlichen Forschung | 149506 | Torsten M Scheyer |
| Schweizerischer Nationalfonds zur Förderung der Wissenschaftlichen Forschung | 162775 | Torsten M Scheyer |
| Generalitat de Catalunya | CERCA Program | Massimo Delfino |
| Agencia Estatal de Investigación | CGL2016-76431-P | Massimo Delfino |
| Horizon 2020 Framework Programme | #695517 | John R Hutchinson |

The funders had no role in study design, data collection and interpretation, or the decision to submit the work for publication.

### Author contributions

Torsten M Scheyer, Conceptualization, Formal analysis, Supervision, Funding acquisition, Investigation, Visualization, Methodology, Writing—original draft, Project administration, Writing—review and editing; John R Hutchinson, Formal analysis, Investigation, Writing—review and editing; Olivier Strauss, Massimo Delfino, Formal analysis, Supervision, Investigation, Visualization, Writing—review and editing; Jorge D Carrillo-Briceño, Rodolfo Sánchez, Resources, Data curation, Validation, Writing—review and editing; Marcelo R Sánchez-Villagra, Conceptualization, Resources, Supervision, Funding acquisition, Project administration, Writing—review and editing

Author ORCIDs

Torsten M Scheyer (iD) https://orcid.org/0000-0002-6301-8983

John R Hutchinson (iD) http://orcid.org/0000-0002-6767-7038

## Decision letter and Author response

Decision letter https://doi.org/10.7554/eLife.49972.sa1

Author response https://doi.org/10.7554/eLife.49972.sa2

## Additional files

### Supplementary files

• Transparent reporting form

### Data availability

All data generated or analysed during this study are included in the manuscript and supporting files. Source data of the Palaeosuchus palpebrosus CT scan dataset can be accessed via CrocBase (https://osf.io/6zamj/).

The following previously published dataset was used:

| Author(s) | Year | Dataset title | Dataset URL | Database and Identifier |
|---|---|---|---|---|
| Hutchinson JR | 2016 | Hatchling Palaeosuchus palpebrosus RVC-JRH-PP1 whole body | https://doi.org/10.17605/OSF.IO/UAD97 | CrocBase, 10.17605/OSF.IO/X38NH |

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
