## [Decision Letter]

**Acceptance summary:**

This manuscript highlights the growing recognition that variation in fossil morphology is common and many taxa break previously proposed constraints on morphology that result in typological thinking. The sacral anatomy of this giant crocodylian is described as comprising three sacrals, breaking the existing constraint that crocodylians have the ancestral condition of only two sacrals, and adding to the list of extinct non-dinosaurian taxa that have more complex pelvic evolution than previously recognized. The authors discuss this in the context of the known effects of *Hox* gene expression shifts, making this a possible example of *Hox* gene-mediated morphological polymorphism that has allowed the adaptation towards the exceptional body size.

**Decision letter after peer review:**

Thank you for submitting your article "Giant extinct caiman breaks constraint on the axial skeleton of extant crocodylians" for consideration by *eLife*. Your article has been reviewed by two peer reviewers, and the evaluation has been overseen by a Reviewing Editor and Diethard Tautz as the Senior Editor. The following individuals involved in review of your submission have agreed to reveal their identity: Michelle Stocker (Reviewer #1); Chris Brochu (Reviewer #2).

The reviewers have discussed the reviews with one another and the Reviewing Editor has drafted this decision to help you prepare a revised submission.

Summary:

The manuscript has received favorable reviews that request only minor revisions, as listed below. Both reviewers agree it is of significance to better understanding Mesozoic crocodylian evolution.

Reviewer #1:

This manuscript highlights the growing recognition that variation in fossil morphology is common and many taxa break previously proposed constraints on morphology that result in typological thinking. The sacral anatomy of *Purussaurus mirandai* is described as comprising three sacrals, breaking the existing constraint that crocodylians have the ancestral condition of only two sacrals, and adding to the list of extinct non-dinosaurian taxa that have more complex pelvic evolution than previously recognized. The authors do a nice job of figuring the relevant anatomy (Figure 4 is especially nice!) and discussing functional and developmental aspects of this morphology. I see no reason that this manuscript cannot be published with a quick revision. Great job!

Reviewer #2:

This is a very interesting manuscript. Admittedly, I was skeptical when I first saw the title. I've seen the odd modern crocodylia specimen with an additional sacral – either a sacralized dorsal or a sacralized caudal – involved in the hip. The authors have done a good job of arguing convincingly that the three-sacral condition is consistent in *P. mirandai* and might have a broader distribution among closely related large-bodied caimanines.

The text is generally clear. There are some grammatical errors, but the points made by the authors were never unclear.

Essential revisions:

- Figure 2: Because the morphology of the ilium is key to your arguments, it would be more effective to separate out H, I, and J as a separate figure (make it 2 and put the pectoral girdle material and reconstruction in a new Figure 3) so as to emphasize its importance and make the parts of the figure larger.

- Subsection “Pelvic girdle”, second paragraph: I would figure this Morphotype 4 so that the reader can compare and contrast for themselves.

- Subsection “Sacral region”: Can you provide any information on the morphology of the centrum faces in addition to the size? Are there flanges on the lateral edges of the centrum faces?

- I may have some photos somewhere of a specimen of *Alligator mississippiensis* with a sacralized dorsal or caudal. The same might be true for other species for which I've seen large samples (e.g. *Crocodylus acutus, Crocodylus niloticus/suchus*). They may still be in 35 mm slide format, but I will look. I'd be happy to share such images with the authors if they're interested.

I'll also check to see if I have any usable *Euthecodon* ilium shots. *Euthecodon* postcranials are few and far between in East African collections, largely because of the way crocodylians have historically been collected in the region. Postcranial material has only recently been collected, and it is generally surface-collected – meaning it often can't really be associated with any particular species. That being said – there's an articulated *Euthecodon* skeleton in the ground at Koobi Fora. I was given some photos of the specimen a few years back, and I can see an ilium. I'll get back with the authors if they're interested in any of this.

- I would recommend dividing Figure 2 into multiple figures. Some of the details salient to the focus of the paper – the sacral rib facets on the ilia in particular – would be much easier to see if they were shown at a larger size. It might be good, for example, to show the pectoral elements, the ilia, and the sacrals as separate figures.

(As an aside on figures, Figure 1 in the version of the manuscript I reviewed was truncated on the right. I suspect this is an artefact of the conversion of the manuscript to PDF format, but it might be good to double-check.)

- A couple of additional fossil caimanine analyses have come out, e.g. Bona et al. on Protocaiman (which the authors cited) and Cossette and Brochu on *Bottosaurus*. I strongly doubt inclusion of such taxa will impact the overall results of this paper, but it might be worth considering the addition of these taxa – especially since the ilium of *Bottosaurus harlani* is known.

(I don't buy the possible synonymy of *M. fisheri* and *M. niger*. I think they're separate taxa and should be treated as such. But again, this wouldn't change anything the authors are saying.)

---

## [Author Response]

Essential revisions:- Figure 2: Because the morphology of the ilium is key to your arguments, it would be more effective to separate out H, I, and J as a separate figure (make it 2 and put the pectoral girdle material and reconstruction in a new Figure 3) so as to emphasize its importance and make the parts of the figure larger.

According to the reviewer’s suggestion, we split the original Figure 2 into a new Figure 2 (pectoral girdle) and Figure 3 (pelvic girdle and sacrum) with enlarged images.

- Subsection “Pelvic girdle”, second paragraph: I would figure this Morphotype 4 so that the reader can compare and contrast for themselves.

A picture and interpretative drawing of the morphotypes 3 and 4 were incorporated in Figure 3—figure supplement 2.

- Subsection “Sacral region”: Can you provide any information on the morphology of the centrum faces in addition to the size? Are there flanges on the lateral edges of the centrum faces?

We have added two sentences to the text under “sacral region” to describe the articulation facets of the sacral centra, especially of the well-preserved primordial sacral 2 of AMU-CURS-541.

- I may have some photos somewhere of a specimen of Alligator mississippiensis with a sacralized dorsal or caudal. The same might be true for other species for which I've seen large samples (e.g. Crocodylus acutus, Crocodylus niloticus/suchus). They may still be in 35 mm slide format, but I will look. I'd be happy to share such images with the authors if they're interested.

We have added another specimen of *Palaeosuchus palpebrosus* to the new Figure 5—figure supplement 1 that shows a pathologic 3-sacral condition. We further thank the reviewer for the offer to provide additional images, but the aim of our paper is not to give a comprehensive overview of developmental aberrations or pathologies in the sacral region in extant crocodylians. We therefore like to refrain from adding more imagery here.

I'll also check to see if I have any usable Euthecodon ilium shots. Euthecodon postcranials are few and far between in East African collections, largely because of the way crocodylians have historically been collected in the region. Postcranial material has only recently been collected, and it is generally surface-collected – meaning it often can't really be associated with any particular species. That being said – there's an articulated Euthecodon skeleton in the ground at Koobi Fora. I was given some photos of the specimen a few years back, and I can see an ilium. I'll get back with the authors if they're interested in any of this.

We thank the reviewer for all the info on *Euthecodon*. As pointed out, most of the material of *Euthecodon* is isolated and the specimen in the field has yet to be described and figured. Therefore, we keep the text as it is at the end of the Discussion.

- I would recommend dividing Figure 2 into multiple figures. Some of the details salient to the focus of the paper – the sacral rib facets on the ilia in particular – would be much easier to see if they were shown at a larger size. It might be good, for example, to show the pectoral elements, the ilia, and the sacrals as separate figures.(As an aside on figures, Figure 1 in the version of the manuscript I reviewed was truncated on the right. I suspect this is an artefact of the conversion of the manuscript to PDF format, but it might be good to double-check.)

According to the reviewer’s suggestion, we split the original Figure 2 into a new Figure 2 and 3 (see comment above). We do not want to split the figures further to comply also with the journal guidelines. The truncation of the image was due to a scaling error that has been fixed in the new version of the Figure 1.

- A couple of additional fossil caimanine analyses have come out, e.g. Bona et al. on Protocaiman (which the authors cited) and Cossette and Brochu on Bottosaurus. I strongly doubt inclusion of such taxa will impact the overall results of this paper, but it might be worth considering the addition of these taxa – especially since the ilium of Bottosaurus harlani is known.

We are aware of the paper by Cossette and Brochu on *Bottosaurus* as well as Souza-Filho et al., 2019 on *Acresuchus pachytemporalis.* The latter does not have the sacral region or an ilium preserved, nor could the morphology and phylogenetic position of the specimen be verified by us so far. We therefore refrain from adding it to the manuscript.

(I don't buy the possible synonymy of M. fisheri and M. niger. I think they're separate taxa and should be treated as such. But again, this wouldn't change anything the authors are saying.)

The reviewer is of course entitled to express their view on this – we refrain from modifying the text and the figure in this regard; as stated in the text *M. fisheri* is a difficult taxon and has been pruned from the analysis shown in Figure 1 accordingly.